EMBO
Molecular Medicine

# Safe CAR-T: shedding light on CAR-related T-cell malignancies

Qibin Liao [1] & Jianqing Xu [2,3,4 ✉]

As of September 30, 2023, the Food and Drug Administration (FDA) Adverse Event Reporting System (FAERS) database has received 12 reports of secondary T-cell malignancies since the first of the BCMA-/CD19-targeted autologous CAR-T therapies was approved in 2017 (https://fis.fda.gov/extensions/FPD-QDE-FAERS/FPD-QDE-FAERS.html). Consequently, FDA has released a statement on November 28, 2023, announcing their ongoing investigation into the identified risk of T-cell malignancy, which has been associated with severe outcomes such as hospitalization and death. On April 18, 2024, the FDA mandated the inclusion of a boxed warning for T-cell malignancies following treatment with BCMA-/CD19-directed autologous CAR-T cell immunotherapies. In this commentary, we have thoroughly elucidated the possible mechanisms underlying theoretical tumorigenesis induced by current viral vectors. Furthermore, we have primarily proposed safer genetic engineering strategies for CAR-T cells, and underscored the necessity of introducing more sensitive and reliable safety evaluation indicators, such as T cell receptor (TCR) diversity and integration site analysis.

See also: P Berg et al

Successful therapeutic outcome of CAR-T relies on stable expression of CAR transgene in T cells, this objective is accomplished through the delivery of CAR transgene utilizing integrating gamma retroviral (RV) or lentiviral (LV) vectors (Table 1). The safety profile some of non-replicating RV/LV vectors has significantly improved by the partial deletion of the 3' long terminal repeat (LTR) U3 region and the substitution of the 5' LTR U3 region with a truncated cytomegalovirus (CMV) promoter for the initial transcription. This approach effectively reduces the transcriptional activity originating from virus LTR (Iwakuma et al, 1999). However, the uncontrolled integration may result in insertional mutagenesis, potentially leading to the upregulation of neighboring oncogenes or the disruption of tumor suppressor genes at the integration site (Modlich et al, 2009). This raises concern regarding the potential tumorigenicity of RV/LV-engineered T cells, although no such observation has been reported to date. However, it is crucial to acknowledge that the uncontrolled integration of viral vectors into the genomes of host cells poses a potential risk of inducing insertional mutagenesis. This has been observed in clinical gene therapy trials for X-linked severe combined immune deficiency (SCID-X1), X-linked chronic granulomatous disease (X-CGD), Wiskott-Aldrich syndrome, and sickle cell disease, where insertional mutagenesis-driven clonal expansion or oncogenesis has been observed in RV/LV-engineered hematopoietic stem cells (HSCs) (Trobridge et al, 2011; Kaiser et al, 2021). Therefore, the theoretical concern of secondary malignancies, particularly with the increasing numbers of CAR-T cell therapies, has consistently accompanied the advancement of all gene therapy products utilizing integrating vectors (Berg et al, 2025). In addition, it should be noted that composite lymphoma (CL), which involves B-cell and T-cell lymphoma, albeit extremely rare, can potentially result in a misleading diagnosis of T-cell lymphoma following CAR-T therapy. Previous investigations have documented instances where individuals initially diagnosed with B-cell lymphoma, who exhibited favorable responses to chemotherapy, were subsequently re-diagnosed with T-cell lymphoma after a period of time (Jin et al, 2023). Furthermore, it is essential to consider pre-existing oncogenic virus infections, such as Epstein-Barr virus, and mutations associated with clonal hematopoiesis, including DNMT3A and TET2 mutations. These factors have been identified as contributors to the development of second T-cell lymphoma following CAR-T therapy, as demonstrated in a recent study (Hamilton et al, 2024). Nevertheless, it is worth noting that the overall benefits of six FDA-approved CAR-T products, including Abecma (idecabtagene vicleucel), Breyanzi (lisocabtagene maraleucel), Carvykti (ciltacabtagene autoleucel), Kymriah (tisagenlecleucel), Tecartus (brexucabtagene autoleucel) and Yescarta (axicabtagene ciloleucel), continue to greatly outweigh their potential risks for their approved uses, because of their remarkable efficacy in treating relapsed or refractory hematological malignancies (Table 1), and these new medicines have saved thousands of lives. Notably, there is currently no direct evidence of viral vector-driven tumorigenesis in secondary malignancies after engineered T-cell therapy, even in T-cell malignancies that were found to be CAR-positive (Jadlowsky et al, 2025; Simon et al, 2023). However, the effort to minimize the potential risks has been persistently explored, including mRNA-based CAR delivery, DNA nanovector-based CAR delivery, CRISPR/Cas-mediated site-specific integration of CAR expression cassette, and nonintegrating lentiviral vectors (NILVs)-mediated CAR-T engineering (Fig. 1).

[1]Department of Oncology and Bio-therapeutic Center, The Third People's Hospital of Shenzhen, Second Hospital Affiliated to Southern University of Science and Technology, 518112 Shenzhen, China. [2]Clinical Center of Biotherapy, Zhongshan Hospital, Fudan University, 200032 Shanghai, China. [3]Institute of Biomedical Sciences, Fudan University, 200032 Shanghai, China. [4]Clinical Center of Biotherapy, Zhongshan Hospital (Xiamen), Fudan University, 361015 Xiamen, China. ✉E-mail: xujianqing@fudan.edu.cn
https://doi.org/10.1038/s44321-025-00205-7 | Published online: 28 February 2025

**Table 1. The FDA-approved CAR-T products.**

| No. | Trade name | Target | Vector | Approval year | Clinical trial | Indication | Efficacy | | | Short-term safety | | Reported T-cell lymphoma cases (Total reported cases)[a] |
|---|---|---|---|---|---|---|---|---|---|---|---|---|
| | | | | | | | Patients | ORR | CR | ≥Grade 3 CRS | ≥Grade 3 NT | |
| 1 | Abecma (idecabtagene vicleucel) | BCMA | LV | 2021 | KarMMa | MM | 128 | 73% | 33% | 5% | 3% | 0 (528) |
| 2 | Breyanzi (lisocabtagene maraleucel) | CD19 | LV | 2021 | TRANSCEND NHL 001 | LBCL | 256 | 73% | 53% | 2% | 10% | 1 (202) |
| 3 | Carvykti (ciltacabtagene autoleucel) | BCMA | LV | 2022 | CARTITUTE-1 | MM | 97 | 97% | 67% | 4% | 9% | 1 (408) |
| 4 | Kymriah (tisagenlecleucel) | CD19 | LV | 2017 | ELIANA | B-ALL | 75 | 81% | 60% | 47% | 13% | 7 (2470) |
| | | | | 2018 | JULIET | DLBCL | 93 | 52% | 40% | 22% | 12% | |
| 5 | Tecartus (brexucabtagene autoleucel) | CD19 | RV | 2020 | ZUMA-2 | MCL | 74 | 93% | 67% | 15% | 31% | 0 (609) |
| | | | | 2021 | ZUMA-3 | B-ALL | 55 | 71% | 56% | 24% | 25% | |
| 6 | Yescarta (axicabtagene ciloleucel) | CD19 | RV | 2017 | ZUMA-1 | LBCL | 101 | 72% | 51% | 13% | 28% | 3 (3729) |
| | | | | 2021 | ZUMA-5 | FL | 104 | 92% | 74% | 7% | 19% | |

*LV* lentivirus, *RV* retrovirus, *MM* multiple myeloma, *LBCL* large B-cell lymphoma, *B-ALL* B-cell acute lymphoblastic leukemia, *DLBCL* diffuse large B-cell lymphoma, *MCL* mantle-cell lymphoma, *FL* follicular lymphoma, *ORR* overall response, *CR* complete response, *CRS* cytokine release syndrome, *NT* neurotoxicity.
[a]Data from the FDA Adverse Event Reporting System (FAERS), accessible through: https://fis.fda.gov/extensions/FPD-QDE-FAERS/FPD-QDE-FAERS.html (Data as of September 30, 2023).

## mRNA-based CAR-T

The majority of therapies utilized in clinical trials and approved CAR-T products in clinical uses are founded on the transduction of RV/LV vectors, resulting in persistent CAR expression. While this approach can yield long-term remissions, it is accompanied by safety concerns, such as cytokine release syndrome (CRS), neurotoxicity (NT), and on-target off-tumor (OTOT) toxicity. Notably, the persistence of CD19-directed CAR-T cells frequently leads to the elimination of CD19-positive healthy B cells, thereby inducing B-cell aplasia and hypogammaglobulinemia, and heightening the susceptibility to infections. Furthermore, the utilization of viral vectors presents several limitations, including restricted capacity for cargo, the requirement for laborious and costly production processes that may result in the emergence of replication-competent viruses, the potential for oncogenic transformation of engineered T cells due to uncontrolled genomic integration, and the risk of immune reactions against vector-derived exogenous agent. mRNA is increasingly recognized as a new category of therapeutic modalities, demonstrating significant potential for mRNA vaccines and various therapeutic applications. The use of mRNA encoding CAR to equip immune cells signifies a developing avenue in cancer and autoimmune disease immunotherapy, facilitating the generation of CAR-positive cells in vitro or in vivo with transient and tunable CAR expression, and mitigating the risk of random transgene insertion. mRNA has been employed in numerous preclinical investigations to introduce CARs into T cells, facilitating their evaluation in model systems for hematological and solid tumors. These studies have demonstrated significant cytotoxicity and tumor ablation (Soundara et al, 2020). Although mRNA-based therapies have exhibited diminished off-target effects, decreased toxicity and alleviated integration-associated safety concerns, the transient expression of protein has posed a drawback in these particular applications. The introduction of CAR constructs into T cells using mRNA has been demonstrated in vitro to have a duration of 7 days, and the limited longevity hinders the ability of engineered T cells to persist, necessitating repeated infusions. However, the persistence of CAR-T in vivo may not be essential for the efficacy of immunotherapy in treating autoimmune disease, which was demonstrated in a clinical trial involving patients with myasthenia gravis who received mRNA-based BCMA CAR-T, where significant decreases in the severity of myasthenia gravis severity were observed for up to 9-month follow-up (Granit et al, 2023).

## DNA nanovector

The use of mRNA transfection for CAR engineering could eliminate the potential risk of insertional mutagenesis. Nonetheless, this approach is impractical for pharmaceutical manufacturing due to the necessity of producing multiple doses per patient, owing to the transient nature of protein expression. T cells, being crucial components of the adaptive immune system, exhibit a high degree of sensitivity toward immunogenic exogenous DNA. Consequently, their functional capacity diminishes swiftly or they trigger apoptosis upon transfection with commonly employed vectors. Therefore, the researchers have developed the nano-S/MARt (nS/MARt) vectors, which have been specifically designed to ensure consistent gene expression while minimizing interference with T-cell functionality. This novel DNA nanovector applies the scaffold/matrix attachment region (S/MAR) motifs to facilitate the episomal maintenance and replication of DNA vectors in actively dividing cells (Bozza et al, 2020). The researchers employed nonintegrating DNA nanovector that exhibited an improved ability to generate genetically engineered cells, and successfully demonstrated its efficient application in human T cells. This DNA nanovector lacks viral constituents and can

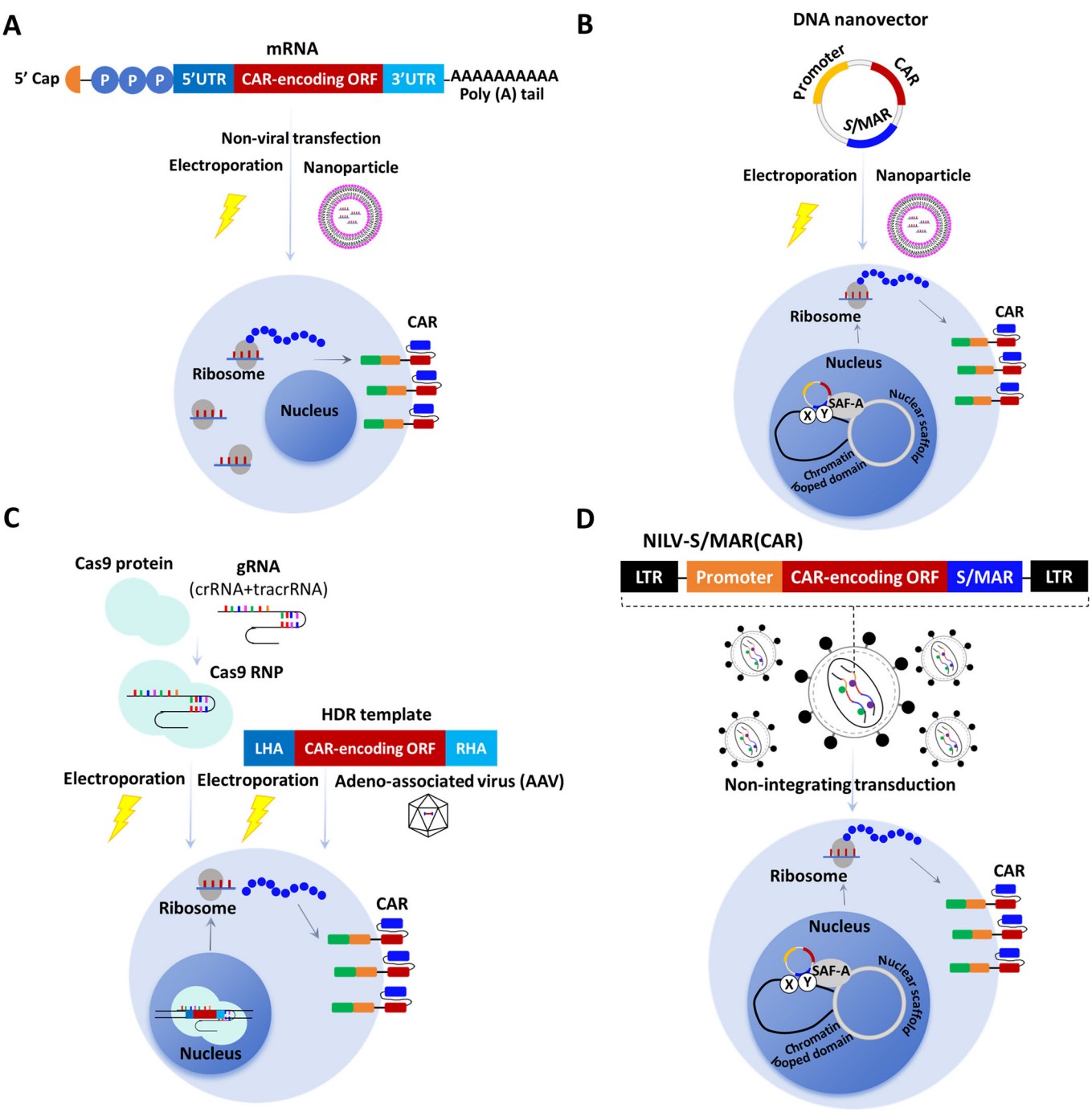

**Figure 1.  Safe CAR-T engineering strategies.**

(**A**) mRNA-based CAR-T. The mRNA-based CAR-T involves the utilization of mRNA with five essential structural elements, a cap structure, a 5′ UTR, the therapeutic CAR-coding ORF, a 3′ UTR, and a poly(A) tail. Before T-cell engineering, the mRNA encoding CAR is transfected in vitro or in vivo by electroporation or nanoparticle, subsequently leading to its translation within the cytoplasm, thereby enabling the transient expression and presentation of CAR on T cells. (**B**) DNA nanovector. An minimally sized DNA nanovector lacking integration capability is comprised of a nuclear S/MAR element and a promoter, enabling persistent expression of CAR transgene, which could be transfected in vitro or in vivo using electroporation or nanoparticle, and the S/MAR element facilitates the attachment of episomal nanovector to the chromosomal scaffold during mitosis, ensuring persistent expression and mitotic stability. (**C**) CRISPR/Cas-mediated site-specific integration. The CRISPR/Cas9 system, comprising an RNP complex composed of Cas9 protein and sgRNA, is delivered into T cells in vitro by electroporation. This is accompanied by the addition of linear dsDNA or AAV carrying an HDR template. The purpose of this process is to achieve site-specific integration of CAR transgene. (**D**) NILVs-based CAR-T. A nonintegrating lentiviral (NILV) vector harboring a S/MAR element, is transduced into T cells either in vitro or in vivo, to facilitate sustained expression of CAR transgenes. UTR untranslated region, ORF open reading frame, S/MAR scaffold/matrix attachment region, sgRNA single guide RNA, RNP ribonucleoprotein, dsDNA double-strand DNA, AAV adeno-associated virus, HDR homology-directed repair, NILV nonintegrating lentiviral vector, SAF-A scaffold attachment factor protein A, X, Y Auxiliary transcription/replication proteins.

replicate extrachromosomally within dividing cells' nuclei, ensuring prolonged transgene expression in T cells without compromising their functionality and genome integrity.

## CRISPR/Cas-mediated site-specific integration

In the field of integrative non-viral methodology, there is a notable surge in technological advancements pertaining to site-specific insertion approaches. This progress can be attributed, in part, to the user-friendly nature of systems like CRISPR/Cas, which have also facilitated early clinical advancements in cancer immunotherapy. These developments hold the potential to address the limitations linked to the utilization of RV/LV and random integration. Notably, a previous study has provided evidence that the electroporation of CRISPR-Cas9 ribonucleoprotein (RNP) and long linear double-strand DNA (dsDNA) template effectively reduces the toxicity of dsDNA template, and further confirms the CRISPR/Cas9 system as an innovative virus-free genome engineering technology in T cells (Roth et al, 2018). Several studies have demonstrated that CRISPR/Cas-mediated genome editing techniques in facilitating the generation of locus-specific integrated CAR-T cells, employing either an adeno-associated virus (AAV) vector or dsDNA as a template. Our Chinese researchers have recently achieved a significant breakthrough by employing CRISPR/Cas9 system to develop a two-in-one methodology for generating non-viral, site-specific integration CAR-T cells. Through the utilization of an optimized protocol, they have successfully demonstrated the feasibility of this approach in a preclinical study by effectively inserting a CD19-directed CAR cassette into the *AAVS1* locus. Moreover, a novel variant of non-viral, PD1-integrated CD19 CAR-T cells was developed and exhibited enhanced capacity in eliminating tumor cells in xenograft models. In the context of adoptive cell therapy for r/r-B-NHL, a substantial proportion (87.5%) of patients achieved complete remission and sustained responses without encountering severe adverse events (Zhang et al, 2022). Taken together, these findings demonstrate the remarkable safety and efficacy of CRISPR/Cas-mediated non-viral, gene-specific integrated CAR-T cells, thereby presenting a groundbreaking

technological advancement in the field of CAR-T therapy. Nonetheless, the induction of chromosomal breaks via the CRISPR/Cas system remains associated with the risk of translocations and potential genotoxicity. This highlights the critical need to consider and monitor the possibility of extensive chromosomal translocations or rearrangements. Furthermore, it necessitates the application of innovative strategies to mitigate some of these genotoxic risks. For instance, employing a combination of different CRISPR nucleases for simultaneous knock-in and base editing in multiplex-edited CAR-T cells represents a promising approach when implementing genome editing in clinical settings (Rayner et al, 2019; Leibowitz et al, 2021; Glaser et al, 2023).

## Nonintegrating lentiviral vectors (NILVs)-based CAR-T

NILVs have been designed with the purpose of mitigating the genotoxicity resulting from insertional mutagenesis by abolishing the integrase activity of LV vectors. Upon transduction with a NILV, cells experience an accumulation of dsDNA circles within nucleus, thereby facilitating the expression of transgene. Nevertheless, this form of episomal DNA undergoes dilution during cell division, leading to the eventual loss of CAR expression in NILV-engineered T cells. In order to address this issue, researchers have made advancements in the development of nonviral plasmids containing a S/MAR element, which allows for long-term expression of transgenes. A novel gene transfer vector called "anchoring nonintegrating lentiviral vector (aniLV)" has been created, which combines the benefits of a S/MAR element and NILV. According to a recent report, compelling evidence was presented to support the notion that an S/MAR-containing NILV has the potential to serve as an ideal vector for the long-term CAR-T engineering, while posing minimal risk of insertional mutagenesis and genotoxicity (Jin et al, 2016).

In summary, the emergence of CAR-T therapies signifies an important advancement in the treatment of hematologic cancers. Although there have been reports of unspecified secondary T-cell malignancies following the widespread use of CAR-T therapy, a standardized methodology for causality assessment is necessary to evaluate pharmacovigilance reports of T-cell malignancies potentially induced by insertional mutagenesis (Berg et al, 2025). In addition, the

development of numerous novel safe CAR-T engineering strategies is expected to enhance safety profiles and circumvent the potential malignant transformation of integrating vectors for patients. Moreover, the present approach employed for determining viral integration-driven carcinogenesis involves evaluating the proliferation of CAR-T cells in the absence of stimulators, including agnostic antibodies and T-cell growth cytokines, which is both time-consuming and lacks sensitivity. To enhance scientific rigor and sensitivity, it may be prudent to consider TCR sequencing at multiple time points prior to, during, and subsequent cultivation to monitor the potential significant amplification of TCR clones. Detection of abnormal TCR clonal expansion during manufacturing should prompt temporary withholding of the cellular product pending completion of insertional mutagenesis risk assessment. Thus, the presence of clonality should be a screening test for infusion. Moreover, TCR diversity can be misleading, as an oligoclonal TCR profile does not inherently signify tumorigenesis. It is essential to include integration site analysis, which provides more pertinent information than TCR diversity alone. However, it is important to establish specific criteria for abnormal amplification values and integration sites in practice, such as defining the normal range, warning range, and abnormal values and integration sites. Improved CAR-T engineering in combination with scientific monitoring methodologies, holds promise in mitigating and eliminate the risk of secondary T-cell malignancies following therapy.

## Peer review information

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

## Acknowledgements

This work is supported by grants from the Shenzhen Science and Technology Program (RCBS20221008093104016), the National Natural Science Foundation of China (82402147), the National Key Research and Development Program of China (2024YFC2311101, 2023YFC2306703, 2022YFC2304401), the Special Funds for Strategic Emerging Industry of Shenzhen (F-2022-Z99-502266), the Shenzhen High-level Hospital Construction Fund (G2022091, XKJS-CRGRK-008) to QL, and the 'Open Competition to Select the Best Candidates' Technology Breakthrough Project for Cell Therapy of NCTIB (NCTIB2023XB01002) to JX.

## Disclosure and competing interests statement

The authors declare no competing interests.

