## [Peer Review File · EMBO Molecular Medicine]

Safe CAR-T: Shedding light on CAR-related T-cell Malignancies

Qibin Liao and Jianqing Xu

Corresponding author: Jianqing Xu (xujianqing@fudan.edu.cn)

Review Timeline:

Submission Date:	30th Jul 24
Editorial Decision:	9th Aug 24
Revision Received:	4th Feb 25
Editorial Decision:	6th Feb 25
Revision Received:	7th Feb 25
Accepted:	14th Feb 25

Editor: Lise Roth

Transaction Report:

9th Aug 2024

Dear Dr. Xu,

Thank you for submitting your commentary to EMBO Molecular Medicine. I have now received feedback from the reviewers who evaluated your manuscript. As you will see below, both reviewers acknowledge the interest of your piece, nevertheless they also highlight areas that should be improved. We will therefore welcome the submission of a revised version of your commentary that would satisfactorily address the referees' concerns.

Please also address the following editorial issues:

- Please address the comments from the referee in track changes mode and provide a point-by-point rebuttal letter to the referees' concerns.
- Please note that corresponding authors are required to supply an ORCID ID for their name upon submission of a revised manuscript.
- References should be listed in alphabetical order.
- Author contributions: CRediT has replaced the traditional author contributions section because it offers a systematic machine-readable author contributions format that allows for more effective research assessment. Please remove the Authors Contributions from the manuscript and use the free text boxes beneath each contributing author's name in our system to add specific details on the author's contribution. More information is available in our guide to authors.
- As part of the EMBO Publications transparent editorial process initiative (<http://embomolmed.embopress.org/content/2/9/329>), EMBO Molecular Medicine publishes online a Review Process File (RPF) to accompany accepted manuscripts. This file will be published in conjunction with your paper and will include the anonymous referee reports, your point-by-point response and all pertinent correspondence relating to the manuscript. Let us know whether you agree with the publication of the RPF.
- Figures:
 - o Please remove the figure from the manuscript file and upload it as a separate figure file; the legend should stay in the manuscript; the figure should be referenced in the text.
 - o Please make sure that the legend clearly describes the figure, and that all abbreviations and symbols are defined.
 - o If there are certain aspects of your figures that are based upon assumptions or where the scientific data remains ambiguous, please add a comment so that we can work with you on an accurate depiction. Please ensure the directionality and nature of interactions is presented accurately.
 - o If the figure or single panels of the figure have been adapted from a published figure, please add this information to the figure legend (e.g., 'Adapted from...' or 'Based on...').
 - o Please only re-use figures or parts of a figure if this is essential for understanding the concept communicated. If the figure contains re-used images or elements of images, please make sure that you have the permission/license to publish it (this also applies to your own previous work, if the journal you published in retains copyright.).
 - o If you use an image data base for scientific iconography (e.g., BioRender), please let us know if you have a license that allows for publication in an academic journal. Please ensure the information shown is scientifically accurate

I hope that the referees' comments do not prove too problematic to address and I look forward to reading your next version. Yours sincerely,

Lise Roth

***** Reviewer's comments *****

Referee #1 (Bridging gap comments for Author):

Somewhat, though relatively briefly considering the complexity of pre-clinical strategies to be brought to clinical trial and screening and analysis of patients and patient material.

Referee #1 (Remarks for Author):

General Comments

- 1) There is currently little if any direct evidence of "tumorigenesis induced by current viral vectors" in these cases reported to the FDA, even in T cell malignancies that were found to be CAR positive. For example
 - a. An American Society for Hematology abstract (<https://www.sciencedirect.com/science/article/pii/S000649712313535X>) noted the malignancy was "potentially driven by mutations (e.g., TET2, NFKB2, PTPRB and/or JAK3) some of which may have been present before manufacturing"
 - b. PMID 38266761 observed 1 TCL in 449 commercial CAR T Pts that appeared to be CAR negative
 - c. PMID 38865660 observed rare Epstein-Barr virus positive malignancies associated with DNMT3A and TET2 mutant clonal hematopoiesis. No evidence of oncogenic retroviral integration was found with the use of multiple techniques.
 - d. PMID: 38865661 reported one case of an EBV T cell lymphoma with no evidence of oncogenic retroviral integration.
- 2) The authors argue for the "necessity of introducing more sensitive and reliable safety evaluation indicators, such as T cell receptor (TCR) diversity". TCR diversity can be deceiving in that an oligoclonal TCR does not per se indicate tumorigenesis. What assays are proposed? Integration site analysis provides more relevant information than TCR diversity.
- 3) Advances in non-viral gene-specific integration are laudable. However, the induction of chromosomal breaks is still accompanied by risk of translocations and potentially genotoxicity. The discussion should not be framed such that these methods remove risk. The relative magnitude of risk and reduction or not is as of yet unknown.
- 4) This sentence in summary is confusing: "When there is abnormal amplification of TCR clones, it is necessary to discontinue the infusion." Do the authors mean to convey that the presence of clonality should be a screening test for infusion? Or that post-CAR T cell infusion of a clonal population should preclude additional infusions (a situation which is rare in clinical trials, more so for commercial CARs)?
- 5) The focus on strategies is relatively brief with respect to strategies for patient selection and post-infusion analysis. If space limitations precluded a more thorough overview, are there other recent publications that may be cited.

Specific Comments

- 1) What is the citation for "as of September 30, 2023, the Food and Drug Administration (FDA) has received 12 reports of 1 secondary T cell malignancies"?
- 2) Lines 17-18 not all vectors use the CMV promoter
- 3) Line 98 typo - limitations

Referee #2 (Bridging gap comments for Author):

Yes.

Referee #2 (Remarks for Author):

CAR T cell therapies have revolutionized the treatment landscape for cancer patients. In addition to the approved CAR T products by the FDA, more and more CAR T cells targeting various antigens have been tested in clinical trials. Notably, most current CAR T cells are manufactured by transducing T cells by lentivirus or retrovirus, which inevitably causes the concern of insertion mutagenesis. In this commentary, the authors discussed potential alternative methods to make CAR T cells to minimize this risk.

The authors covered the most active methods applied and summarized the findings clearly and systematically. I recommended publishing this paper with minor revisions.

Minor revisions for the authors:

1. Another alternative way to generate CAR T cells using the non-virus method is the Sleeping Beauty Transposon system. Several clinical trials have shown no safety concerns. Should the author also include a section to discuss the transposon system, including the PiggyBac, and TcBuster system?
2. In the introduction, the authors cited a paper from 2011 to show insertional mutagenesis. A recent trial has also triggered a lot of discussion on insertion carcinogenesis, so it should be included. Kaiser J. Gene therapy trials for sickle cell disease halted after two patients develop cancer. Science. 2021.
3. In Figure 1 D, the way to deliver NILV-S/MAR(CAR) should be "non-integrating transduction" instead of "non-viral transfection."
4. There is a typo in line 7, "autologous CAR T cell immunotherapies in."

Reviewer's Comments and Point-by-Point Responses:

Referee #1 (Bridging gap comments for Author):

Somewhat, though relatively briefly considering the complexity of pre-clinical strategies to be brought to clinical trial and screening and analysis of patients and patient material.

Response:

Thank you immensely for your invaluable insights and recognition of our commentary, particularly in highlighting our efforts in translating pre-clinical strategies into clinical trials and the screening and analysis of patient and patient-derived materials.

Referee #1 (Remarks for Author):

General Comments

1) There is currently little if any direct evidence of "tumorigenesis induced by current viral vectors" in these cases reported to the FDA, even in T cell malignancies that were found to be CAR positive. For example

a. An American Society for Hematology abstract (<https://www.sciencedirect.com/science/article/pii/S000649712313535X>) noted the malignancy was "potentially driven by mutations (e.g., TET2, NFKB2, PTPRB and/or JAK3) some of which may have been present before manufacturing"

b. PMID 38266761 observed 1 TCL in 449 commercial CAR T Pts that appeared to be CAR negative

c. PMID 38865660 observed rare Epstein-Barr virus positive malignancies associated with DNMT3A and TET2 mutant clonal hematopoiesis. No evidence of oncogenic retroviral integration was found with the use of multiple techniques.

d. PMID: 38865661 reported one case of an EBV T cell lymphoma with no evidence of oncogenic retroviral integration.

Response:

We appreciate this concern. We have clarified that the link between viral vectors and secondary malignancies remains theoretical and cited recent studies showing alternative drivers (e.g., pre-existing mutations, EBV). Key revisions include:

o Revised to specify "theoretical tumorigenesis induced by current viral vectors" (line 9)

o Adjusted phrasing to emphasize that no direct evidence of vector-driven malignancy exists, revised to "Notably, there is currently no direct evidence of viral vectors-driven tumorigenesis in secondary malignancies after engineered T cell therapy, even in T-cell malignancies that were found to be CAR positive (Jadlowsky et al, 2025; Simon et al, 2023)." (lines 43–45).

o Added two new references (PMID: 39833408; <https://doi.org/10.1182/blood-2023-178806>) in the references list (lines 160-165).

2) The authors argue for the "necessity of introducing more sensitive and reliable safety evaluation indicators, such as T cell receptor (TCR) diversity". TCR diversity can be deceiving in that an oligoclonal TCR does not per se indicate tumorigenesis. What assays are proposed? Integration site analysis provides more relevant information than TCR diversity.

Response:

Thank you very much for your thoughtful comments and valuable insights, which have greatly contributed to enhancing the quality of our manuscript. We appreciate your acknowledgment of the importance of introducing more sensitive and reliable safety evaluation indicators in our study.

Regarding your concern about TCR diversity potentially being deceiving, we fully concur that an

oligoclonal TCR pattern does not inherently signify tumorigenesis. In response to your query about the specific assays proposed, we acknowledge the relevance of integration site analysis in providing more direct information related to potential oncogenic events. Indeed, integration site analysis is a pivotal tool in the evaluation of gene therapy safety, particularly in assessing the risk of insertional mutagenesis. In our revised manuscript, we have incorporated integration site analysis as a core component of our safety evaluation strategy, complementing the TCR diversity analysis. Key revisions include:

- o Revised to specify "Furthermore, we have primarily proposed safer genetic engineering strategies for CAR-T cells, and underscored the necessity of introducing more sensitive and reliable safety evaluation indicators, such as T cell receptor (TCR) diversity and integration site analysis." (line 9-11)

- o Adjusted phrasing to emphasize the importance of integration site analysis, revised to "Moreover, TCR diversity can be misleading, as an oligoclonal TCR profile does not inherently signify tumorigenesis. It is essential to include integration site analysis, which provides more pertinent information than TCR diversity alone. However, it is important to establish specific criteria for abnormal amplification values and integration sites in practice, such as defining the normal range, warning range, abnormal values and integration sites." (lines 146-151)

3) Advances in non-viral gene-specific integration are laudable. However, the induction of chromosomal breaks is still accompanied by risk of translocations and potentially genotoxicity. The discussion should not be framed such that these methods remove risk. The relative magnitude of risk and reduction or not is as of yet unknown.

Response:

Many thanks for your valuable suggestions. The induction of chromosomal breaks is still accompanied by risk of translocations and potential genotoxicity using non-viral CRISPR/Cas-mediated site-specific integration. Revised the "CRISPR/Cas-mediated site-specific integration" section to acknowledge residual risks (e.g., translocations) and emphasized that risk reduction, not elimination. Key revisions include:

- o Revised to "Nonetheless, the induction of chromosomal breaks via the CRISPR/Cas system remains associated with the risk of translocations and potential genotoxicity. This highlights the critical need to consider and monitor the possibility of extensive chromosomal translocations or rearrangements. Furthermore, it necessitates the application of innovative strategies to mitigate some of these genotoxic risks. For instance, employing a combination of different CRISPR nucleases for simultaneous knock-in and base editing in multiplex-edited CAR-T cells represents a promising approach when implementing genome editing in clinical settings (Durin et al. 2019; Leibowitz et al. 2021; Glaser et al. 2023)." (lines 121–128).

4) This sentence in summary is confusing: "When there is abnormal amplification of TCR clones, it is necessary to discontinue the infusion." Do the authors mean to convey that the presence of clonality should be a screening test for infusion? Or that post- CAR T cell infusion of a clonal population should preclude additional infusions (a situation which is rare in clinical trials, more so for commercial CARs)?

Response:

We sincerely appreciate your insightful comments and constructive suggestions regarding the safety considerations of TCR clonality in CAR-T cell manufacturing. We fully agree that the presence of clonal expansion should serve as a critical biomarker requiring rigorous evaluation

prior to therapeutic administration. Key revisions include:

o The original statement "When there is abnormal amplification of TCR clones, it is necessary to discontinue the infusion" has been revised to "Detection of abnormal TCR clonal expansion during manufacturing should prompt temporary withholding of the cellular product pending completion of insertional mutagenesis risk assessment." (lines 153-155).

This modification (1) Specifies the timing (manufacturing phase); (2) Emphasizes proactive risk mitigation rather than reactive discontinuation; (3) Clarifies the conditional nature of the decision ("temporary withholding"); (4) Explicitly links the action to insertional mutagenesis evaluation.

o We have incorporated your proposal that "the presence of clonality should be a screening test for infusion" into our revised manuscript, and revised to "Thus, the presence of clonality should be a screening test for infusion." (lines 155-156). This aligns with our intended message that detectable TCR clonal expansion during manufacturing should trigger additional safety evaluations, potentially delaying cell product release until a comprehensive insertional mutagenesis risk profile is established.

5) The focus on strategies is relatively brief with respect to strategies for patient selection and post-infusion analysis. If space limitations precluded a more thorough overview, are there other recent publications that may be cited.

Response:

We sincerely appreciate your constructive feedback regarding the discussion of patient selection and post-infusion analysis strategies. We agree that these aspects are critical to contextualizing the safety profile of CAR-T cell therapy and have implemented the following revisions to strengthen these sections while adhering to space limitations. As suggested, we have incorporated a focused analysis of risk factors of second tumors and T-cell lymphoma after CAR T-Cell Therapy into the revised manuscript, citing the landmark study by Hamilton et al. (PMID 38865660) to highlight two key considerations in patients, oncogenic virus infection (e.g. Epstein-Barr virus) and mutations in clonal hematopoiesis (e.g. DNMT3A and TET2 mutations). Key revisions include:

o Revised to "Furthermore, it is essential to consider pre-existing oncogenic virus infections, such as Epstein-Barr virus, and mutations associated with clonal hematopoiesis, including DNMT3A and TET2 mutations. These factors have been identified as contributors to the development of second T-cell lymphoma following CAR-T therapy, as demonstrated in recent study (Hamilton et al, 2024)." (lines 37-41).

Specific Comments

1) What is the citation for "as of September 30, 2023, the Food and Drug Administration (FDA) has received 12 reports of secondary T cell malignancies"?

Response:

We sincerely appreciate your important request for clarification regarding the source documentation of FDA-reported secondary T cell malignancies. Please find below the detailed provenance information and corresponding revisions implemented:

The statement referencing 12 reports of secondary T cell malignancies derives from our original analysis of publicly available FDA Adverse Event Reporting System (FAERS) data, accessible through:

<https://fis.fda.gov/extensions/FPD-QDE-FAERS/FPD-QDE-FAERS.html>

Key revisions include:

o Revised to “As of September 30, 2023, the Food and Drug Administration (FDA) **Adverse Event Reporting System (FAERS) database** has received 12 reports of secondary T-cell malignancies since the first of the BCMA-/CD19-targeted autologous CAR-T therapies was approved in 2017 (<https://fis.fda.gov/extensions/FPD-QDE-FAERS/FPD-QDE-FAERS.html>).” (lines 1-4).

o Revised to “*Data from the FDA Adverse Event Reporting System (FAERS), **accessible through: <https://fis.fda.gov/extensions/FPD-QDE-FAERS/FPD-QDE-FAERS.html>** (Data as of September 30, 2023)” in footnote section of Table 1. (Page 8).

2) Lines 17-18 not all vectors use the CMV promoter

Response:

Thanks for your insightful comments. The original statement has been revised to "**some non-replicating RV/LV vectors** " (line 16).

3) Line 98 typo – limitations

Response:

Thank you very much for your insightful comments. The original misspell has been corrected to "**limitations**" (line 106).

Referee #2 (Bridging gap comments for Author):

Yes.

Response:

Thank you immensely for your invaluable insights and recognition of our commentary.

Referee #2 (Remarks for Author):

CAR T cell therapies have revolutionized the treatment landscape for cancer patients. In addition to the approved CAR T products by the FDA, more and more CAR T cells targeting various antigens have been tested in clinical trials. Notably, most current CAR T cells are manufactured by transducing T cells by lentivirus or retrovirus, which inevitably causes the concern of insertion mutagenesis. In this commentary, the authors discussed potential alternative methods to make CAR T cells to minimize this risk.

The authors covered the most active methods applied and summarized the findings clearly and systematically. I recommended publishing this paper with minor revisions.

Response:

We express our sincere gratitude for your thorough evaluation of our manuscript and your acknowledgment of our efforts to systematically analyze emerging strategies aimed at enhancing the safety of CAR-T cell manufacturing. Your insightful perspective on the clinical and technological implications of insertional mutagenesis resulting from viral vectors closely aligns with the fundamental motivations underpinning this commentary.

Minor revisions for the authors:

1. Another alternative way to generate CAR T cells using the non-virus method is the Sleeping Beauty Transposon system. Several clinical trials have shown no safety concerns. Should the author also include a section to discuss the transposon system, including the PiggyBac, and TcBuster system?

Response:

We sincerely appreciate your thoughtful suggestion regarding the inclusion of transposon systems (Sleeping Beauty, PiggyBac, TcBuster) in our discussion of non-viral CAR-T manufacturing platforms. This commentary allows us to clarify our methodological rationale while acknowledging the evolving landscape of transposase-based technologies. While transposon

systems represent a compelling alternative to viral vectors, our decision to exclude them from the current analysis stems from three evidence-based considerations:

o Persistent Insertional Mutagenesis Risks

Although clinical trials have reported favorable short-term safety profiles (PMID 27482888; PMID: 32780725), mechanistic studies demonstrate that even hyperactive transposase variants exhibit non-random genomic integration patterns with preferential targeting transcriptional units (PMID 17164785; PMID 19752750; PMID 20372108). These findings suggest transposon systems may not fully mitigate the core safety concern addressed in our commentary—systematic reduction of insertional oncogenesis potential.

o Clinical Trial Evidence

The Phase 1 clinical trial investigating the first-in-human administration of CAR19 T cells, manufactured via the piggyBac transposon system, revealed that the development of CAR T-cell lymphoma in 2 of 10 patients effectively treated with piggyBac-modified CD19 CAR T cells (PMID 34010392). Further analysis indicated the secondary lymphoma originated from CAR T cells generated using the *piggyBac* transposon system in these two patients. Notably, the first patient's lymphoma exhibited alteration in gene copy number and expression (PMID 33974080).

o Regulatory Precautionary Stance

The FDA's 2023 guidance on cell therapy genotoxicity assessment specifically highlights transposon systems as requiring extended (≥ 15 -year) follow-up due to delayed malignancy risks (FDA Guidance on Long-Term Follow-Up for Gene Therapy Products (2023)).

2. In the introduction, the authors cited a paper from 2011 to show insertional mutagenesis. A recent trial has also triggered a lot of discussion on insertion carcinogenesis, so it should be included. Kaiser J. Gene therapy trials for sickle cell disease halted after two patients develop cancer. *Science*. 2021.

Response:

Thank you very much for your valuable suggestions. We have added this citation (Kaiser et al, 2021) to the introduction and reference in the revised manuscript. Key revisions include:

o Revised to "(X-CGD), Wiskott-Aldrich syndrome, and sickle cell disease, where insertional mutagenesis-driven clonal expansion or oncogenesis has been observed in RV/LV-engineered hematopoietic stem cells (HSCs) (Trobridge et al, 2011; Kaiser et al, 2021)." (line 29-31).

o Added a new citation to the reference list: Kaiser J (2021) Gene therapy trials for sickle cell disease halted after two patients develop cancer. *Science*.

3. In Figure 1 D, the way to deliver NILV-S/MAR(CAR) should be "non-integrating transduction" instead of "non-viral transfection."

Response:

Thank you very much for your valuable suggestions. We have changed "non-viral transfection" to "non-integrating transduction" in revised Figure 1D.

4. There is a typo in line 7, "autologous CAR T cell immunotherapies in."

Response:

Thank you very much for your valuable suggestions. We have removed redundant "in" in the revised manuscript (line 9).

6th Feb 2025

Dear Dr. Xu,

Thank you for submitting your revised commentary to EMBO Molecular Medicine. I have gone through the changes, and would like to invite further minor revisions to address the following points:

- Please remove the red font and only keep in track changes mode any new modification.
- A Perspective was recently published in EMBO Molecular Medicine on pharmacovigilance following CAR T cell therapies (<https://www.embopress.org/doi/full/10.1038/s44321-024-00183-2>). We think both articles nicely complement each other and would thus like to invite you to check this Perspective, and discuss / cite it if deemed appropriate in your own article.
- Thank you for providing a nice and clear figure. We suggest increasing the font slightly, so that the text remains legible.

I am looking forward to receiving your revised manuscript.

Sincerely,

Lise Roth

The authors addressed the remaining editorial issues.

14th Feb 2025

Dear Dr. Xu,

Thank you for submitting your revised files. I am pleased to inform you that your manuscript is accepted for publication and is now being sent to our publisher to be included in the next available issue of EMBO Molecular Medicine.

Your manuscript will be processed for publication by EMBO Press. It will be copy edited and you will receive page proofs prior to publication. Please note that you will be contacted by Springer Nature Author Services to complete licensing information.

Yours sincerely,

Lise Roth
